# Cluster-Based Multi-User Multi-Server Caching Mechanism in Beyond 5G/6G MEC

**DOI:** 10.3390/s23020996

**Published:** 2023-01-15

**Authors:** Rasha Samir, Hadia El-Hennawy, Hesham Elbadawy

**Affiliations:** 1Department of Electronics and Communications, Faculty of Engineering, Ain Shams University, Cairo 11566, Egypt; 2Network Planning Department, National Telecommunications Institute, Cairo 11768, Egypt

**Keywords:** B5G, 6G, CMUMS caching, MEC

## Abstract

The work on perfecting the rapid proliferation of wireless technologies resulted in the development of wireless modeling standards, protocols, and control of wireless manipulators. Several mobile communication technology applications in different fields are dramatically revolutionized to deliver more value at less cost. Multiple-access Edge Computing (MEC) offers excellent advantages for Beyond 5G (B5G) and Sixth-Generation (6G) networks, reducing latency and bandwidth usage while increasing the capability of the edge to deliver multiple services to end users in real time. We propose a Cluster-based Multi-User Multi-Server (CMUMS) caching algorithm to optimize the MEC content caching mechanism and control the distribution of high-popular tasks. As part of our work, we address the problem of integer optimization of the content that will be cached and the list of hosting servers. Therefore, a higher direct hit rate will be achieved, a lower indirect hit rate will be achieved, and the overall time delay will be reduced. As a result of the implementation of this system model, maximum utilization of resources and development of a completely new level of services and innovative approaches will be possible.

## 1. Introduction

The increasing interest in taking mobile communication technology and its applications to life and business has dramatically revolutionized and created a new digital world with unleashed values and low costs. Back in the mid-1990s, mobile technology was obscure, but it has become a necessity everywhere because of its broad functionality. Mobile communication technology started by supporting mobile data connectivity in addition to voice-based communications, moved to High-Definition (HD) video conferencing and then to reality. Mobile communication technology began as a stunning achievement in the world of technology, but it is now evolving to be a more sustainable human-centric technology. Technology innovations such as the Internet of Things (IoT), Software-Defined Networking (SDN), and 6G are transforming the development of software applications across several industries [1,2,3]. Delivering content to users and delivering real-time data to centralized data centers requires massive amounts of near-real-time computation. For example, the IoT system can benefit from the powerful B5G/6G MEC [4]. It rapidly transfers data and real-time monitoring information of physical objects to a fully digital world or virtual environment through Virtual Reality (VR), Augmented Reality (AR), Mixed Reality (MR), Extended Reality (XR), or any other Metaverse-related technologies while orchestrating real-time information exchange between the physical and digital worlds [5,6]. Using real-time video conferencing for e-learning and online meetings helped us a lot during the COVID-19 pandemic and lockdown. The evolution of mobile communication technology has streamlined our lives and saved us both time and resources.

MEC 6G network innovation solution empowers 6G mobile network capabilities by consolidating compute, database, storage, and other cloud services closer to the edge of the network at the speed of 6G [7,8]. It enables low-latency communication for gaming, real-time collaboration, and hosting other emerging services. In addition, moving cloud computing services to the edge of B5G and 6G networks and handling large amounts of data close to their source significantly reduces the load on the mobile backhaul. MEC reduces application costs by operating it at the border of the mobile network without the need to utilize costly backhaul links [9]. It also allows storage and data stream processing at the border instead of being completed at the data center core or remote cloud environments. MEC’s ultra-low latency and local data processing capabilities meet some sensitive and private security requirements [10]. MEC’s ultra-low-latency solution accelerates data flow and delivers real-time information collection for various applications [11]. There are multiple factors that should be considered such as the infrastructure of the edge, compute applications, the support of real-time communications, the desired time delay, server capacity, and population coverage [12,13].

The tremendous and rapid growth of mobile communication technologies and their applications, especially the imposing demand on video content delivery, has added a huge load on the mobile backhaul uplinks between the Base Stations (BSs) and core network, and also on the servers at the core of the networks, which can be optimized by reducing the physical path between the User Equipment (UE) and the cloud computing servers [14]. In addition, modern application quality constraints of minimal time delay, jitter, and packet loss have added more complexity to the mobile network design strategy [15]. The MEC architecture allows heavy traffic loads to be handled without causing network bottlenecks while delivering application quality.

MEC is introducing a compelling and encouraging model to mount caching capabilities in adjacency with the Beyond 5G/6G network base stations, and it is providing rapid, popular content delivery of today’s delay-sensitive applications at the speed of mobile networks backhaul [16]. Content caching is typically deployed at the core of wireless networks [10,17]. Edge servers and caching techniques are deployed at the edge of the mobile network [18]. By installing servers and content caching close to the end users, communication between them will be faster, the response time will be shorter, and the traffic load on the core network will be reduced. Conducting a cached copy of the content from the network edge to users saves a significant delay in passing through the core uplinks toward the central servers with every request for the same content. The goal of content caching in edge computing is to preserve a copy of the content at the network edge on the path between the end user and its central servers [19]. Besides caching content near end users, MEC also controls the selection metrics of the cached content, as well as how to maintain the cached content [20]. The selection metrics define the content that should be cached when to update the cached content, and for how long the cached copy should be kept. Thus, the content of high prospects is cached. The content of low prospects is not cached in order to improve the content caching solution.

In this study, we have studied the model described in [21], and we have worked on optimizing the server’s content caching mechanism so that it guarantees the redundancy of high-demand tasks while maintaining cache replacement behavior in order to improve performance. Furthermore, with the use of bonded clusters of servers, MD and MEC servers within a local domain can communicate and coordinate task handling. The outcomes are reduced communication times and increased direct hits. In addition, agent servers facilitate communication between local domain clusters, as well as communication with remote cloud environments. We introduced the CMUMS caching algorithm to optimize the content caching mechanism and control the replication of highly requested content. It considers the modern application demands for ultra-low latency, the end-user device capability such as mobility and storage limitations, along with content hit rates, value, and size. A key contribution of our paper is a solution to the caching problem in a layout where caching components are distributed with finite backhaul and storage capabilities. The main objective of this study is to develop and apply the MEC caching system model. This will shed light on the parameters of the system and how they could be used to assess performance.

The remainder of the paper is organized as follows: Section 2 represents the related work, Section 3 discusses the discussed system model, the proposed system description with system model parameters and proposed system model formatting of mathematical Components, Section 4 illustrates the results and analysis, and, finally, the paper is concluded in Section 5.

## 2. Related Work

A significant amount of research has been conducted recently on the advantages of computing and cache offloading in MEC systems. MEC content caching strategy, resource allocation, and computational offloading decisions have been examined in many studies. However, in order to determine the most appropriate approach to content caching, we need to understand that there is still no single methodology that satisfies all the disparate end-user needs and fulfills all application requirements. Researchers and engineers have been focusing on studying and sharing the MEC key features, opportunities, and challenges. With the help of categorizing each opportunity and challenge details, researchers can select the techniques that are most appropriate for their applications, which is beneficial for research and future work in the area. Furthermore, the classification emphasizes bibliometric overviews for ease of selecting algorithms and for future research, enabling researchers to focus on applications and further research, as well as addressing the gaps in this area, which also identify the open challenges and future research directions.

The survey [5] presents a technology overview and application scenarios for future MEC technologies. In addition, a discussion of how MEC is converged with 6G, AI, blockchain, and the metaverse. Several unresolved problems in MEC are also summarized. In [7], the survey study, a comprehensive perspective, was adopted to highlight the importance of architectural transformation to 6G. It was also adopted to highlight the appropriate architectural improvements that can be made to wireless future technologies. These improvements can be made in order to achieve the benefits of enhanced spectrum utilization, increased data rates, reduced latency, and reduced CAPAX and OPEX. The survey [8] reviews key research areas, technological advancements, and areas of future research in MEC Beyond 5G/6G. The survey [13] examines the security, dependability, and performance aspects of 5G MECs. A brief overview of background knowledge on MEC is provided by referring to current standardization efforts. It also explains each aspect of 5G MEC by introducing related terms, recent developments, and challenges. However, the challenges associated with B5G/6G MEC require further investigation. Ref. [22] presents the importance of securing 5G-MEC from end-to-end against known vulnerabilities, threats, and attacks. It also highlights the necessity of unifying the 5G-MEC security framework. The author in [23] describes the most significant factors for developing B5G/6G edge computing capabilities and applications in various industry sectors. The author [24] discusses the characteristics and major challenges of the proposed three frameworks for incorporating satellite and MEC systems. He also presents relevant research concerns for the development of a highly intelligent, highly secure, and essentially unified sustainable network. However, practical lab tests are required to investigate the proposed structures. A 6G MEC network-based methodology described in [4] addresses challenges related to the Web of Things System (WoT) and Microservices-compliant Load Balancing (McLB) in urban mashups. However, further research into more advanced scenarios is still needed to improve performance. Ref. [25] provides an overview of the numerous opportunities and challenges in AI-based 6G security and privacy. Additionally, it also identifies further research opportunities for challenges in AI-based security and privacy provision as well as feasible options to address these challenges. The author of [26] discussed an open-source MEC architecture for 6G networks based on innovative decoupling and redesign techniques. A comprehensive survey [27] reviews the benefits and challenges of integrating network green communication strategies and MEC caching methods on public health and environmental stability. Ref. [28] presents a survey of modern inventions in MEC and Caching in servers; it also presents several motivations for open research. In the survey [29], the algorithms have been classified into eight distinct classes, and each class is explained based on its respective merits and demerits. In [30], the author provides a survey of the relevant literature on research motivations and challenges to using MEC systems for a variety of services to enable user mobility. A detailed survey is conducted in [31] to uncover and evaluate the features and improvements in task offloading and content caching methods in MEC. Research directions are presented for device-enhanced MEC and the key challenges. Optimization of content caching has been studied in multiple works; a Software-Defined Networking (SDN) approach has been considered for mobile networks. The clustering-based approach is also examined to reduce the estimated delay and enhance overall solution performance.

Caching configuration and performance should be controlled by implementing an appropriate content caching approach. This should take into account all the application characteristics and features in the given scenario. The author in [15] presents a detailed review of RL-enabled MEC and outlines the use cases of RL algorithms in addition to exploring practical guidelines for further investigation. The author in [32] proposed optimization to the MEC-enabled Internet of Vehicles (IoV) system, where the MEC server serves as an Anchor Node (AN) that handles computing tasks for Electric Vehicle Nodes (EVNs) as well as transmitting energy via Simultaneous Wireless Information and Power Transfer (SWIPT) to it. However, the discussed offloading technique does not take into account the finite capacity and utilization of the MEC server. The author [1] presents a time-division model for developing communication and MEC networks for delay-sensitive IoT based on the process-oriented concept. However, a UAV’s payload setup should be properly considered in order to maximize its effectiveness in resource use. The author [2] proposed a radio map (RM) design and deployment approach using deep reinforcement learning (DRL) in MEC networks including the proposed joint offloading and resource allocation algorithm for efficient handling of computational offloading.

The author in [12] proposed a Comp-HO computational handoff algorithm that considers the quality of the network signal and the workload of MEC servers. However, the study focused primarily on Mobile Augmented Reality (MAR) use cases. The system model needs to take into account both the increased energy consumption at high handover volumes as well as a load of exchanged control plane messages. A scalable, hierarchical caching policy [18] that is aware of the rate of access and data volume is proposed. However, it is based only on file size and popularity. It did not take the server utilization or caching redundancy into account. There is a slight efficiency drop initially after implementing the policy until statistics are collected for each object. In [33], the author proposed multiple revised methods to improve network energy efficiency and delay reduction for heterogeneous 5G MEC networks. However, no definitive solution is provided. The paper [34] presents a cluster-based solution for cooperative MEC content caching. However, it studied only video files with known sizes and popularity that remain the same for some time without considering cluster formation or communication within the cluster. The offloading mechanism in [35] aims to lower the required energy and the time delay that users will experience in Non-Orthogonal Multiple Access NOMA-MEC networks. Still, there are some improvements that need to be made. These improvements should take into account the popularity of the tasks as well as the energy-saving aspects on the server side. In [6], the author proposed combining VR video rendering and caching at one MEC node, as well as the cooperative sharing of caches among multiple MECs. The author in [23] describes the most significant factors for developing B5G/6G edge computing capabilities and applications in various industry sectors. Another study [36] discussed the secure caching mechanism. This paper proposed a model for a cloud-to-edge server communication model for 6G wireless-mobile technology and optimized internet web navigation and file sharing. The author in [37] proposed a two-step optimal solution that integrates task cache update into computation offloading. This assumption assumes that Orthogonal Frequency Division Multiple Access (OFDMA) is the network technology. Ref. [38] studied applying the Deep Q-learning Network (DQN) algorithm to increase the hit ratio and minimize time delay. However, it is assumed that all contents have an equal size (2000 bits) and servers’ caching capacity is sufficient to carry all of the favored contents. The author [39] proposed an extensive iterative neural network design for building an intelligent cache system. This model calculates content ranking, the geolocation of the user, and subsequent content requests. However, if the estimation is not correct, the Base Station (BS) will have to fetch the requested content from the central content server and then forward it to the users, which increases latency. Ref. [40] proposes a modification of reinforcement learning (RL) caching methods to enhance network and user performance and customize MEC servers. However, it does not have a centralized management plane, nor does it cooperate with other servers. In [41], the author discusses IoT caching in opaque mobile networks; the process predicts the next location of the user using 6-tuple parameters and forwards the content request to the backend where it is cached. However, it relies on having the GPS location service enabled.

The discussed model in [21] proposes the Multi-User Multi-Server Caching (MUMSC) algorithm for MEC content caching. Nevertheless, the used task scoring model was limited to task popularity and size; it did not take into consideration the server’s maximum caching capacity, queue utilization, server structure formation, used a sub-optimal path between the Mobile Device (MD) and the caching server via a proxy server, and it did not guarantee redundancy for highly demanding tasks on all servers. A priority-based offloading and caching mechanism is presented in [42] for offloading and caching according to the local computing strategy. In spite of this, the allocation of resources did not take into account energy use and cooperative formation. Ref. [43] reports on a study that increased the Average Downloading Percentage (ADP) for vehicle-to-telecom network connectivity (LTE-V2I). However, the proposed caching strategy should consider more parameters such as content size, cache size, energy consumption, and cluster formation. Cluster formation in [44] decreases cache redundancy. It analyzes the historical statistics of the number of requests in order to estimate future probabilities of those requests. However, the process of cluster creation is not discussed. The proposed content caching strategy in [45] is Multi-agent Reinforcement Learning (MARL)-based to improve the hit ratio and minimize the time delay. However, the size of the Q-table is large. In [46], a model for forecasting placement based on Short-term Time-Varying (STV) quality is proposed. However, only live streaming was considered for caching.

Our previous work in [9] focused on the orchestration of MEC computation jobs as we proposed a cluster-based energy-aware offloading framework that automates the collaboration between the MEC edge servers and orchestrates the offloading of computation efficiently. The perspective of the concurrent paper is completely different from our previous work in [9] as we are studying a new cluster-based MEC caching system model. In our previous work [9], we introduced a solution to orchestrate the computation jobs within the Beyond 5G/6G MEC architecture using the proposed Iterative System Algorithm (ISA). In this paper, our work main objective is to investigate the designed framework, address the integer optimization problem, and optimize the MEC content caching technique for B5G/6G network edges. Our proposed work is unique in considering modern algorithms according to the best of our knowledge, which are not covered yet in the literature. Moreover, the chosen literature of the studied researchers is classified in the matter of performance benchmarks by describing the field of improving performance and the areas where the opportunities for development stand as a future study by further researchers.

## 3. System Model

The discussed model in [21] introduced four application, communication, and caching system structures for discussion and reference. It consists of a group of MEC servers with enabled computation tasks and caching services, a central agent server, and a connection to the main cloud servers. The MEC servers in each structure topology are distributed and surrounded randomly by multiple MDs within a particular coverage area. The MD is communicating with collective MEC servers within the coverage area, and the MDs are requesting various computation tasks, as illustrated in Figure 1. The MEC cache storage space is limited. The number of jobs to be offloaded or cached, as well as the number of computation tasks, will be limited due to the finite resources of the server. The server can store multiple tasks, while a task result might be cached on all or only one server according to the model scoring criteria. Caching capability is the same on all MEC servers, while it can be easily expanded to variable caching spaces.

MD is always communicating with the surrounding servers to obtain the task result. It is assumed that MD computation requests can be processed directly by MEC servers when the task has already been cached. The central proxy/agent server is contacted by the edge server when the required task is not cached on any of the local servers; the agent server will then forward the task to the server located in the remote cloud to be processed. Thus, the time delay is increased, and the main core uplinks are used.

By analyzing the model in [21], several factors were not taken into consideration, including the design of the topology of the servers, the server caching capacity limit, queue usage, and the lack of redundancy for high-demand tasks. Our work endeavors to optimize the server’s content caching mechanism, and it is designed to control the redundancy of high-demand tasks while maintaining cache replacement behavior for better results. The creation of bonded clusters of servers allows MD and MEC servers within the local domain to communicate and orchestrate the handling of tasks, thereby increasing direct hit rates and reducing time delays. The agent server enables the communication between different local domain servers, and it also facilitates communication with remote cloud environments.

### 3.1. Proposed System Model

Our work presented in Figure 2 is proposing an efficient Cluster-based Multi-user Multi-Server (CMUMS) caching strategy to enhance the performance in beyond a 5G/6G MEC based environment. Our work is compliant with the 3GPP TS 23.501 (version 17.4.0) [10]. Initially, our work started with validating the discussed caching mechanisms in [21] that introduces two caching mechanisms, Task Popular Caching (TPC) and Multi-User Multi-Server task Caching (MUMSC). In TPC, the selective caching scheme occurs based on task popularity, so not all the task results will be cached. In MUMSC, the caching selection criteria consider the task popularity and the task size. It has also considered two benchmarks: Optimal Task Caching (OPTC) and Remote Cloud Server Response (RCSR). The best-case scenario OPTC assumes that the requested task result by MDs is already cached in their closest and frequently reached MEC servers. Thus, the task result will be fetched directly from the edge with minimum latency, which requires huge storage space. The worst-case scenario RCSR assumes that the MEC servers are not caching the contents, and the MD requests are sent directly to the remote servers located at the main data center.

In the discussed model, task size and popularity were the only factors considered in the scoring model. There was inadequate consideration of the maximum caching capacity of the server, queue utilization, server structure formation, the use of a suboptimal path between the Mobile Device (MD) and the caching server via the proxy server, and there was no redundancy for the highly demanding tasks on all servers. Our work introduces an efficient CMUMS caching algorithm. The main approach in designing the cluster is to share the load of incoming requests, enhance the task request distribution, enable redundancy, and deploy modified MUMSC caching methodology as a new edge caching solution that satisfies today’s application requirements of ultra-low latency, reduces the energy consumption, and improves the end-user QoE, while offloading the core servers and uplink.

### 3.2. Proposed System Description

In our proposed caching system model, the MEC servers are created in a clustered environment and distributed in hexagonal topology formation. The clusters are connected and controlled by agent/proxy servers. The cluster design considers our CMUMS model, server utilization, in addition to enhancing the task distribution while maintaining the task utilization and redundancy. The proposed topology structures are plotted as in the following Figure 3 (the proposed topology structures are also tested using Unity tool as shown later). The four different sizes of the structures as our baseline are 500 × 500, 700 × 700, 1000 × 1000, and 1500 × 1500 m. In each topology structure, a group of MEC servers is distributed in hexagonal shapes as an enhancement to the random MEC servers distribution in [21]. They are surrounded randomly by multiple MDs within a particular coverage area. It was considered that there would be 10 MDs and 5 MEC servers in topology-1, 20 MDs and 10 MEC servers in topology-2, 30 MDs and 15 MEC servers in topology-3, and 40 MDs and 20 MEC servers in topology-4. The proposed algorithm introduces an optimization problem of integer programming to present the task status. In such type of programming, the result is zero or one only. The zero value means that the task is not cached, and the value of one means that it is cached. When the requested computational task is already cached, the MD will obtain the task result from the directly connected server in a very short time, eliminating the additional time or delay to obtain the task result from a remote central server, and accelerates the data flow with real-time data processing and provides much ultra-low latency.

Our proposed CMUMS caching algorithm aims to optimize the servers content caching mechanism and control replicating the high-probability demanding tasks. It considers the interest of the task result from end-users in order to improve the storage capability and control the caching redundancy. The task result attributes define if the task should be cached or not cached. The high probability demanding tasks are cached into multiple distributed servers to enhance the end-user QoE by reducing the latency and increasing the direct hit rate. In the same manner that the MEC server may cache multiple different task results, the task result might be cached and replicated into multiple MEC servers. We evaluate caching the tasks as per the proposed scoring criteria that considers the task popularity, task size, and task replication with respect to the maximum caching space Q as presented in Algorithm 1. Our model is proposed to control the redundancy of the high-probability demanding tasks while maintaining the storage cache replacement behavior for the new task results. We assumed reserving the same percentage of storage space in all MEC servers for replicating the high-probability demanding tasks. We studied preserving up to 20–25% from the Q size. We found that our CMUMS increases the direct hit rate in comparison to MUMSc that has higher indirect hit rate than direct hit rate. Tasks are cached based on our proposed intelligence scoring criteria. It considers the servers where the most tasks are requested and which tasks have high probability. The reservation percentage of the Q is adaptive and can be changed due to the total caching space. As a model, MEC servers’ content caching should be more selective and determine what task results to be cached and for how long in order to optimize the storage capability and the caching redundancy. Concerning the task attributes, the higher probability requested tasks have a higher chance of being stored on more than one server within the cluster in a particular location. Our work addresses the integer optimization problem that controls which task will be cached and the hosting servers list which minimizes the servers’ communication time. Building the servers in bonded clusters allowed for orchestrating the communication between the servers in the local domain, cluster, and also between the MD and its directly connected cluster servers. In the manner, the agent server is enabling the communication between the different servers within the same local domain, and it also enables the communication with the remote cloud environments. Maintaining a dedicated storage percentage on all servers for caching the highly demanding tasks offered advantages such as minimizing the delay and increasing the direct hit rates. The task attributes are considered in scoring the tasks and managing the caching process for the remaining storage percentage, and also optimizing the Q utilization. We also studied dividing the large task results into smaller parts and storing it in multiple servers while the agent server is maintaining a storage mapping database pointing to the part hosting servers.

Our algorithm consists of a set of MEC servers S with caching space Q, assuming that servers receive a set of tasks K. The task scoring algorithm considers the tasks’ attributes such as size and popularity in determining whether the task will be cached or not. Each MEC server is surrounded by a set of randomly distributed MDs. The MEC servers are sorted in ascending order of their distance from the center of the cluster which means that the smaller the number, the shorter the range. For redundancy, we specify a capacity percentage from the caching total capacity Q; this specified space has a predetermined threshold V. Once a task is received, the server checks if the task should be cached or not according to the described caching criteria that is illustrated in the flow chart in Figure 4.

We assumed the (V) is same in all servers, and we evaluated the solution when (V) equals 20–25%. We refer to the utilized storage out of (V) as Psum value, and it equals zero before receiving any task. The remaining storage space percentage is $Q_{j}$, which equals around 75–80% in our study; we refer to the remaining current storage space as $C_{j}$ and the remaining available storage space is calculated as $C_{k}=Q_{j}-C_{j}$. For each received highly demanding task, the server compares the task result size with the available space of (V). Once the task result size is less than the available (V) space, the task will be cached in all servers as $x_{j}^{k}$ = 1. For the tasks that should be cached in all servers, the task result will be added to Psum and Psum will be increased by k size. When a task is received and the (V) space is not enough to accommodate the task result, the task will not be cached into all servers. It will be only cached in the nearest server with enough $Q_{j}$ storage. The proxy server checks all the connected severs $Q_{j}$ storage availability starting from the first connected server in order to cache the task in the nearest server with enough storage. When the received task that should be cached in (V) space while its result size is larger than the $C_{k}$, it will be fragmented and its fragments will be stored in multiple servers with reference. Caching the highly ranked tasks within the cluster makes it reachable to the MDs with minimum delay and increases the direct hit rate.
**Algorithm 1** The proposed algorithm for Cluster-based Multi-user Multi-Server Caching mechanism in B5G/6G MEC.       **Input**      1:Set of MEC servers S      2:Set of tasks K      3:Cashing Space of MEC servers Q      4:Threshold V      5:Size of computational task K      6:The ratio of task result to the whole size of task K      7:The popularity of task K      8:The distance of each server j to the center of the cluster             **Output**      9:List $x_{j}^{k}$ where
$x_{j}^{k}$ = 0                                                               ▷ Not cached (Default value)$x_{j}^{k}$ = 1                                                                        ▷ Cached on MEC server         **Algorithm**    10:Sort the MEC servers in ascending order of their distance from the center of the cluster.    11:Sort tasks in a descending order according to the ratio of the task size to the task popularity.    12:$P_{sum}$ =0      k = 0    13:**while**$P_{sum}$< V     **do**    14:    $P_{sum}$ += size of k    15:    $x_{j}^{k}$ =1 ∀ j    16:    k++    17:**end while**    18:**while** k < K     **do**    19:    j = 0    20:    **while** j < size of S     **do**    21:        **if** size of k < $Q_{j}-C_{j}$    22:        $x_{j}^{k}$ = 1        and **break**    23:        **else**    24:            j++    25:    **end while**    26:        k++    27:**end while**    28:j = 0 k = 0    29:**while**$Q_{j}\neq C_{j}$     **do**    30:    **while** k < K     **do**    31:        **if** $x_{j}^{k}\neq 1$∀ j     **then**    32:           Save tasks on multiple servers    33:           $C_{k}$ = 0    34:           **while** $C_{k}$< K     **do**    35:               **if** $Q_{j}\neq C_{j}$ **then**    36:                   $C_{k}$ += $Q_{j}-C_{j}$    37:                   $x_{j}^{k}$ = 1    38:               **end if**    39:               j++    40:           **end while**    41:        **end if**    42:        k++    43:    **end while**    44:**end while**

### 3.3. Proposed System Model Formatting of Mathematical Components

The task response time is the total delay until the MD obtains the task result. We assumed that all MEC servers have enough hardware configuration to handle the allocated load with ultra-low processing time to neglect the server processing time from the total delay calculation. To solve the optimization problem, the terms are discussed in the following equations with the same list of symbols and acronyms in Table 1 are used to simplify the comparison.

Suppose that the transmit power $p^{t}$ between the MEC server and MD is set to 20 mWatt. We considered channel power gain $h^{t}$ as in Equation (Equation 1) regarding [21], and we considered it again equal to Equation (Equation 2) as per the value in 3GPP TS 23.501 (version 17.4.0) [10].

Channel power gain as in [21]
(1)$$h^{t}=127+30*log\left(d\right)$$The channel power gain may be expressed as follows, which is matching the 3GPP 5G propagation model [10]. The center frequency $\left(f_{c}\right)$ is normalized by 1 GHz, and *d* is the distance between the communicator:(2)$$h^{t}=32.4+20log\left(f_{c}\right)+30log\left(d\right)$$

To calculate the uplink data flow speed between the $MD_{i}$ and the MEC server *j* by deploying the Shannon–Hartley, the result formula is in Equation (Equation 3):(3)$$r_{i,j}=Blog_{2}\left(1+\frac{p^{t}h^{t}}{\sigma }\right)$$
where the system bandwidth *B* is 20 MHZ, and the noise power σ is $4\times 10^{-3}$ mWatt at the receiver.

When the $MD_{i}$ has a task *k* request, the system will behave in one of three possible scenarios:The first scenario (Directly-Connected Edge Caching): when the $MD_{i}$ communicates with the directly connected MEC servers and check for a cached copy of the task result with one of the possible following cases:
–$x_{j}^{k}$ = 1, and only one MEC server *j* has the required task result, then $MD_{i}$ obtains the task result direct from the server *j*.–$x_{j}^{k}$ = 1 and the task *k* result is cached in multiple servers, then $MD_{i}$ prefers to obtain the task result from nearest server and the task *k*
The total delay is calculated as in Equation (Equation 4):
(4)$$t_{i,j}^{k}=\alpha_{i,j}\frac{\rho_{k}\beta_{k}}{r_{i,j}}.$$The second scenario (Indirectly-Connected Edge Caching): when there is no directly connected server has a cached copy of the requested task *k*, the agent server starts communicating with the indirectly connected server $j^{\prime }$ to the $MD_{i}$, and it looks for a cached copy of the task result. If available, the agent server obtains the task result from the edge server and transfers it to the $MD_{i}$ so the task *k* total delay is a bit longer and is calculated as in Equation (Equation 5):
(5)$$t_{i,a,j^{\prime }}^{k}=\alpha_{i,j^{\prime }}\left(\frac{\rho_{k}\beta_{k}}{r_{a,j^{\prime }}}+\frac{\rho_{k}\beta_{k}}{r_{i,a}}\right)$$The third scenario (Central Cloud Caching): when there none of the edge servers honors a cached copy of the task *k* result, $MD_{i}$ will be redirected to communicate with the central server at the remote cloud network, and the task *k* total delay is calculated as in Equation (Equation 6).The time delay to obtain the task *k* result from the main data centers is high. Over and above that might occur at the same time to numerous of computation tasks, the task *k* total delay is the highest in comparison to the scenarios of caching the task result at MEC servers, i.e., $t_{i,c}^{k}$ is higher than $t_{i,\alpha ,j^{\prime }}^{k}$ and $t_{i,j}^{k}$:
(6)$$t_{i,c}^{k}=\alpha_{i,j}\frac{\rho_{k}\beta_{k}}{r_{i,c}}$$

Caching Decision: the MEC servers are caching the entire application content and its relevant data as task caching. The MD obtains the task result rapidly from the closest MEC server with the cached task. In addition to the probability of caching the task result in MEC server or not (1 or 0) as in Equation (Equation 7), the task result could be cached in more than one MEC server. We consider the task popularity, the task size, and the caching space utilization as task attributes:(7)$$X=\left\{\begin{array}{cc} 0, & \mathrm{server}\;j\;\mathrm{doesn}’t\;\mathrm{cache}\;\mathrm{computation}\;\mathrm{task}\;k\\ 1, & \mathrm{server}\;j\;\mathrm{caches}\;\mathrm{computation}\;\mathrm{task}\;k \end{array}\right.$$
where X = $x_{j}^{k}:j\in S,k\in K.$

The total delay to transmit the task from the server to $MD_{i}$ is $T_{i}$ and is calculated as in Equation ([Disp-formula FD8a-sensors-23-00996]); it is a variable delay since the $MD_{i}$ can obtain the task result in different scenarios as follows:
(8a)$$T_{i}=\alpha_{i,j}[T_{i\psi }{|}_{x=1,0,0}+T_{i\phi }{|}_{x=0.1,0}+T_{i\zeta }{|}_{x=0,0,1}]$$The requested task result is cached in the directly connected MEC edge server, and the $MD_{i}$ obtains it directly with minimum delay as in Equation ([Disp-formula FD8b-sensors-23-00996]):
(8b)$$T_{i\psi }=x_{j}^{k}log(1+\frac{t_{i,j}^{k}}{t_{dmax}})$$The requested task result is not cached in any of the directly connected MEC edge servers and the $MD_{i}$ obtains it from indirectly connected edge server via the proxy/agent server, and the delay $T_{i\phi }$ is calculated as in Equation ([Disp-formula FD8c-sensors-23-00996]):
(8c)$$T_{i\phi }\;=(x_{j^{\prime }}^{k}log(1+\frac{t_{i,a,j^{\prime }}^{k}}{t_{dmax}})+\left(1-x_{j^{\prime }}^{k}\right)log(1+\frac{t_{max}}{t_{dmax}}))$$The requested task result is not yet cached in any of the edge servers while it is available in the central cloud servers; the $MD_{i}$ obtains it from the central cloud via the proxy/agent server, and the $T_{i\zeta }$ is the largest delay in comparison to the previous scenarios, and it is calculated as in Equation ([Disp-formula FD8d-sensors-23-00996]):
(8d)$$T_{i\zeta }=log(1+\frac{t_{i,c}^{k}}{t_{dmax}})$$

There are also two persistent parameters $t_{dmax}$ and $t_{max}$: $t_{dmax}$ is applied to manage the variation in response delay results when task sizes are unequal, while $t_{max}$ prevents the error method from choosing the longer delay and $\alpha_{i,j}$ is the connection probability between mobile devices *i* and server *j*.

The optimization problem aims to minimize the communication time delay by allowing the frequently accessed tasks to be cached on all or a group of servers based on the proposed algorithm as per Equation (Equation 9). Consequently, the received task request by any MD has a great probability to be found on one of the communicated MEC, which, in turn, minimize the demand to acquire the result from the agent server or any other server:(9)$$\begin{array}{cc} P: & \min_{x_{j}^{k},j^{\prime }\in S_{i}}\sum_{i=1}^{n}T_{i} \end{array}$$
s.t.
$$\begin{array}{cc} \; & \;\;\;\;\;\;\sum_{k=1}^{K}x_{j}^{k}\rho_{k}\leq Q_{j}\;\;\;\;\;\forall j\in S\\ \; & \;\;\;\;\;\;x_{j}^{k}\in 0,1,\;\;\;\;\;\;\;\;\forall j\in S,k\in K \end{array}$$When solving the problem, the main objective is to determine what task should be cached and where to cache it as the Q size is limited according to the mentioned constraints.

## 4. Results and Analysis

Our experimental setup consists of computer-based MATLAB 2016 simulation software running on top of Intel i7 Quad-core 3.2 GHz CPU and 64 GB RAM to evaluate the proposed model parameters. We have also used real-time developed tool using Unity version 2020.3.24F1 LTS to measure the coverage statistics. To analyze the impact of deploying the cluster formation and the CMUMS mechanism, we measured the impact of deploying the proposed mechanism on four different scalable hexagonal clustered topologies. We considered 10 MDs and 5 MEC servers in topology-1, 20 MDs and 10 MEC servers in topology-2, 30 MDs and 15 MEC servers in topology-3, while only 40 MDs and 20 MEC servers in topology-4. The system construction of each considered topology is presented in Figure 3. We also developed a Unity real-time tool as presented in Figure 5 to plan and design the MEC server allocation, to ensure that we have proper coverage and performance, and to evaluate the correct deployment and desired operation of the proposed architecture. The tool allows full topology mapping that is defined by space size, number of MDs and MEC servers plus its location. We measured the solution effectiveness on different topology structures with the following sizes: 500 × 500, 700 × 700, 1000 × 1000 and 1500 × 1500 m. The system applied parameter values are presented in Table 2.

We considered four different topology sizes with different popularity factor α values. We measured the effect on the time delay in the different topologies at caching space Q = 100 MB and 250 MB. Our results are compared with MUMSC, TPC, OPTC and RCSR as referenced in [21]. In TPC, the major factor that affects the task caching is the task popularity; the MEC server caches the computing task with the highest popularity until there is no more space to cache other tasks. In the RCSR case, the tasks are not cached and the MDs obtain the task results from the central cloud server. In OPTC, it assumed that all tasks are cached in the edge servers, and the task result is obtained from the connected servers with the minimum delay. Results in Figure 6 present the advantage of using the CMUMS caching algorithm on minimizing the time delay versus average task sizes from 1 to 10 MB. We have started our lab tests with an estimated workload of 1000 tasks. This is accomplished using the MATLAB built-in function of random generation by Gaussian distributions. The average task size ranges from 1 to 10 MB. By using the MATLAB imported function Zipf distribution for generating random requests, 120,000 requests are estimated to have been generated. The same assumption has been applied to the four MEC topology structures, and the results have been analyzed. Zipf distribution is applied to complete the task popularity with respect to the distribution and task redundancy design. Zipf distribution is commonly used to analyze the task popularity distribution. In this method, the tasks are distributed in an exponentially decaying that will help to represent the highly popular tasks at the beginning of the curve with less popular tasks at the curve tail. The steep of the curve is controlled by a factor named Zipf parameter α. The task results with high popularity have a great opportunity to be cashed on all servers, and the delay is calculated for each average task size.

The average task size is calculated by generating tasks on Gaussian distribution that changes from 1 MB to 10 MB in step 1 MB with different average value. The average task size does not indicate the actual task size, and it indicates where the tasks are distributed. Thus, each task size is considered and formulated as an array with the same size. Consider that $\rho_{k}$ and $\beta_{k}$ are generated randomly based on Gaussian distribution with (mean, variance) equal to (10, 2) MB and (0.5, 0.15). To obtain the task result, a time delay is needed. The relationship between the delay and task size is presented in Figure 6 at Q = 100 MB, and the delay differences are noticeable when the variation of the average task size values is high. The CMUMS reduces the delay time compared to MUMSC for the same environmental topology. i.e., CMUMS delay time is 0.15 s at the average task size of 1 MB, delay is almost 0.55 s at the average task size of 10 MB, while MUMSC delay time is 0.25 s at the average task size of 1 MB, and delay is almost 0.65 s at the average task size of 10 MB.

As shown in the previous Figure 6, for MUMSC [21] in topology 3, at Q = 100 MB, the time delay is around 0.7 sec, while in CMUMS, it is around 0.6 sec, which means that the time delay is minimized by at least 10%. Therefore, a cross-section is presented to illustrate this impact in Figure 7. With the increase of Q, the percentage of cached tasks is increased and hence the time delay needed to obtain the task result will be minimized. To illustrate the positive impact on minimizing the delay, a cross-section is presented in Figure 8 for Q values up to 250 MB.

To deeply discuss the effect of the CMUMS algorithm on minimizing the time delay, we check the required time delay to obtain the task result as per CMUMS and compare the result when the same task result is requested in the other referenced caching methodology presented in [21]. The presented values in Table 3 show that, with the increasing of α, our CMUMS decreases the time delay. The cluster formation in the proposed CMUMS structure (with the distribution and redundancy mechanisms to caching the tasks) minimizes the time delay needed to obtain the result of the requested task. It saved around 30% compared to the MUMSC in our lab test. The highlighted columns start from the discussed α values in different scenarios starting with the value in [21] and other values with step 0.1 as an example.

As observed, the CMUMS cluster formation and the task redundancy design enhancements can considerably promote the caching hit ratio and decrease downloading the content from the center cloud server (which means the task is cached in one of the MECs in the cluster or via agent/proxy server) when compared with the referenced algorithms in [21]. Moreover, the proposed CMUMS also optimizes the overall delay as the requested tasks will be almost found within the clusters; hence, the delay is minimized. In the following comparisons, we explore the direct and indirect hit rate, server occupancy and caching capacity at different α values. The assumed average task size is 10 MB. Topology 4 is chosen as a case study to illustrate the following comparisons. We also discuss the CMUMS algorithm behavior at different popularity factor α. The α value indicates the steep of the curve and hence affects on the number of highly popular tasks to be cached and on which servers. i.e., tasks with small popularity may not be cached or ignored. In Figure 9, we measure the direct hit rate versus caching maximum capacity Q at different Zipf α values 0.56, 0.66, 0.76 and 0.86.

To demonstrate the effect of the CMUMS on the direct hit rate at Q = 250 MB and different α values, the tasks are distributed and cached according to the proposed scoring criteria. The area under the curve (that represents the high popularity tasks) increases as the Zipf parameter value gets higher. As discussed in CMUMS caching mechanism, with respect to the Q size, the high frequently accessed tasks are the most cached and replicated into the MEC edge servers within the cluster. Table 4 shows the enhancements of more than 65% in the direct hit rate in CMUMS compared to the referenced algorithms in [21].

Once the MEC server receives a task request, it will check if the result of the requested task is already cached in the local storage. If the task is not in the local server storage, the server will communicate with the proxy/agent server to learn about the nearest edge server that caches the requested task result and communicates with it in order to obtain the requested result. In case none of the edge servers is caching the task result, the proxy/agent will then communicate with the remote central cloud server to obtain the task result, and the edge server will then forward it to the requester $MD_{i}$ and also check if the obtained result should be cached in the local server storage.

The controlled edge server cluster formation and task distribution decrease the indirect hit rate as the highly popular tasks are cached on all edge servers up to the reserved storage space (V). While the other tasks will be cached partially in some edge servers, the proxy/agent server maintains the server to task result caching mapping database to obtain the requested result from the nearest edge server before considering the remote cloud servers. Figure 10 shows how the indirect hit rate is being affected compared with the direct hit rate, where, if the task is found on one of the directly connected MECs, the direct hit rate will increase in opposition with the indirect hit rate for the same instant. Table 5 presents the comparison between the referenced algorithms in [21] CMUMS when *Q* = 250 MB. The indirect hit rate is minimized by a saving ratio of 35% which positively affects the time delay required to obtain the task result, and that also minimizes the server utilization ratio needed to handle the received tasks.

We studied the direct and indirect hit rate behavior at different Q values up to 250 MB where topology 4 is selected at α = 0.76 with average task size equals 10 MB for a set of K tasks, the preserved threshold V is set to 25%, and the considered channel power gain $h^{t}$ equals Equation (Equation 2) as per [10]. We found that the direct and indirect hit rate behavior may follow one of the highlighted three regions as in Figure 11.
Region 1: in this region, the curve is slightly fluctuated as the Q value is close to the assumed average task size. Hence, the efficiency of the existing redundancy is not noticeable. In real scenarios, the fluctuated portion will almost decay/disappear as Q value in real scenarios will be much higher than the average task size in our lab test.Region 2: at certain Q values, the redundancy effectiveness threshold is considered a starting point to represent the enhancements of CMUMS. It illustrates an incremental behavior in direct hit rate as in Figure 11a in contrast to the decremental behavior in indirect hit rate as in Figure 11b compared to the referenced algorithms in [21].Region 3: at higher Q values, the opportunity to cache more highly demanding tasks is better. The direct hit rate is also increased. When the Q is larger at higher α, the tail of the curve has a lot of mapped tasks with almost the same popularity.

As illustrated in Figure 12, the direct hit rate increases when the requested task is cached on the directly connected servers. Enabling the caching redundancy on the edge servers according to the configured percentage (V) guarantees to reduce the latency in access to the resources. In Figure 12a region 1 at Q = 40 MB, we noticed a slight decrease in the direct hit rate because of demanding some task results that are not available on the directly connected servers. Accordingly, as in Figure 12b, the indirect hit rate increases when the requested task results are available on indirectly connected edge servers. Despite the decrease in direct hitting and the increase in indirect hitting in some instances, the time delay to obtain the task result is still lower than obtaining the task results from the remote cloud servers. There will be no hits when there is no edge server honoring a cached copy of the required task result, and the task result is downloaded from the central cloud servers.

Consequently, orchestrating the task distribution between the edge servers and maintaining a mapping database at the proxy/agent server is enhancing the capability to obtain the requested task result fast. It is also optimizing the edge servers processing utilization, and decreasing the need to utilize the core links towards the central cloud servers as shown in Figure 13.

A close cross section in Figure 14 is to present the effect on the server utilization ratio versus the caching capacity at different α values. The server utilization at Q = 250 MB is nearly 1.89 in TPC and MUMSC, while the utilization ration at the same Q is around 1.67 in CMUMS. The CMUMS saving ratio in server utilization is around 18–20%.

In the study of the MUMSC [21], topology 4 is selected with 20 MEC servers, and the tasks are distributed between the edge servers based on the defined scoring criteria. We found that some servers are overutilized when handling the task requests such as the servers with index number 5, 10, 17, and 18 during the peaks, while the other servers are underutilized as observed in Figure 15. The edge servers utilization is variable because of the lack of controlling the tasks’ redundancy. The discussed caching model in [21] considered the popularity and task size as the scoring attributes. The introduced task distribution mechanism did not ensure avoiding the situation where some edge servers are overutilized while other edge servers are underutilized.

Our proposed CMUMS caching criteria include task redundancy and controlling the task distribution in the cluster formatted structure to achieve a better server load sharing as presented in Figure 16.

The homogeneity in the heatmaps in Figure 16 is clear due to the existence of the redundant distribution and handling of tasks between the cluster members compared to the greedy distribution in Figure 15. Our results present that our proposed solution will strongly calculate the nearest caching node to obtain optimal performance in terms of outcomes compared with classic caching distribution strategies. Importantly, the results are showing that the proposed algorithm can lower the cost in an outstanding manner when storage capacity is limited in addition to reducing the time delay to obtain the task result, increasing the direct hit rate, and enlarging the task result redundancy within the cluster while maintaining the MEC servers processing and storage load sharing. The main outcomes of the presented work may be summarized in the following Table 6.

Our proposed CMUMS model reduced the time delay by 30%, reduced the indirect hit rate by 35%, and enhanced the direct hit rate by 65% compared to the referenced MUMSC model [21] in our study analysis.

## 5. Conclusions

Caching at the edge is one of the MEC’s optimistic solutions that fulfill the modern QoS requirements as it brings the critical contents close to the mobile end-users. It brings highly demanding content to the mobile BSs to eliminate the overhead of duplicating the communications with the core servers to download the same copy. Thus, MEC offers the distinctive opportunity to work on enhancing the architectural design and deployment strategy of content caching and is not limited to just edge caching. Eliminating the possible core network congestion while mitigating the utilization of the backhaul uplinks encourages the work more on developing context-aware edge caching architecture design, big data analysis techniques for MEC, and cost-saving distributed caching methodologies. Deploying content caching at the edge of beyond 5G and 6G MEC networks enables providing rapid access to highly popular content with ultra-low latency and mitigates the limited backhaul uplinks capacity that requires a convenient strategy of edge caching. Optimizing the content caching addresses the problems of network optimization in addition to the related end-user QoE.

Based on the findings of our paper, it can be concluded that the demonstrated and proposed model and its CMUMS caching algorithm, cluster build, task distribution, and the role of the agent/proxy server outperform the other discussed models from different perspectives. With respect to the previously published works, our proposed system model has improved the overall direct hit rate by around 65 percent, whereas the indirect hitting has been reduced by more than 30 percent. Moreover, the overall time delay has been reduced by more than 30 percent. Tasks with high popularity will be cached on all the cluster servers. The presented work enlarges the task result redundancy within the cluster while enhancing the overall performance by maintaining the MEC servers’ processing, storage load sharing, energy, and cost. In addition, the overall system is improved by eliminating more than 90 percent of the ongoing traffic inside the network from being served by the remote cloud servers. The use of edge computing in mobile technology will always be subject to challenges that can be categorized as environmental challenges, such as temperature, humidity, and the use of renewable energy sources; energy generated by renewable energy sources occurs at certain times of the day. System challenges include inadequacies in CPU, memory, storage, and battery storage. Network challenges such as available bandwidth, latency, and security. In fact, developing the mobile edge caching constructive methodology and making it a more advantageous and cost-saving solution became more important. This requires more orientation, thorough vision, application selection, and further consideration, moreover, for the following investigations and development work.

## Figures and Tables

**Figure 1 sensors-23-00996-f001:**
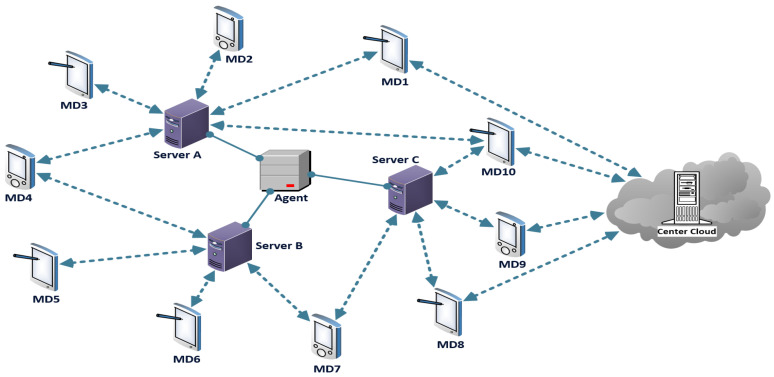
The discussed system model design.

**Figure 2 sensors-23-00996-f002:**
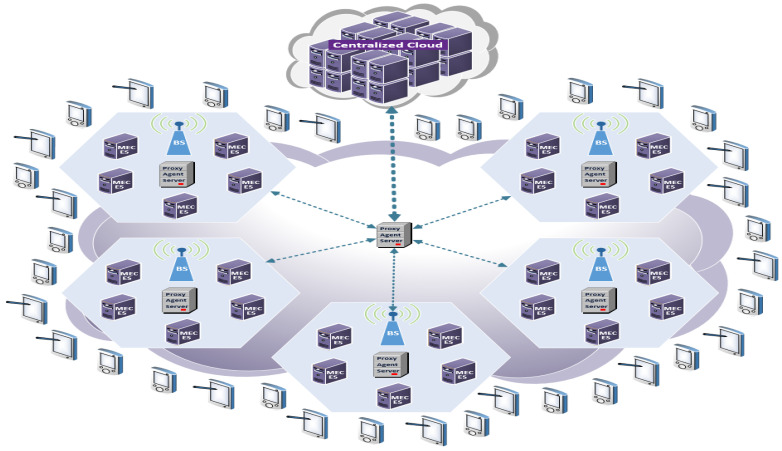
The proposed system model for the CMUMS Caching mechanism.

**Figure 3 sensors-23-00996-f003:**
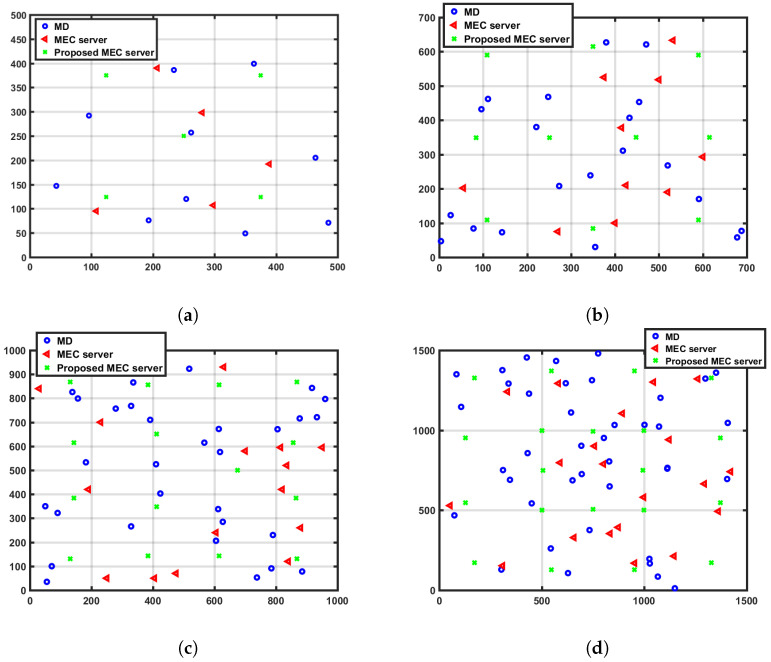
The proposed distribution of MECs in four different topology structures. (**a**) First Topology Structure in 500 × 500 m; (**b**) Second Topology Structure in 700 × 700 m; (**c**) Third Topology Structure in 1000 × 1000 m; (**d**) Fourth Topology Structure in 1500 × 1500 m.

**Figure 4 sensors-23-00996-f004:**
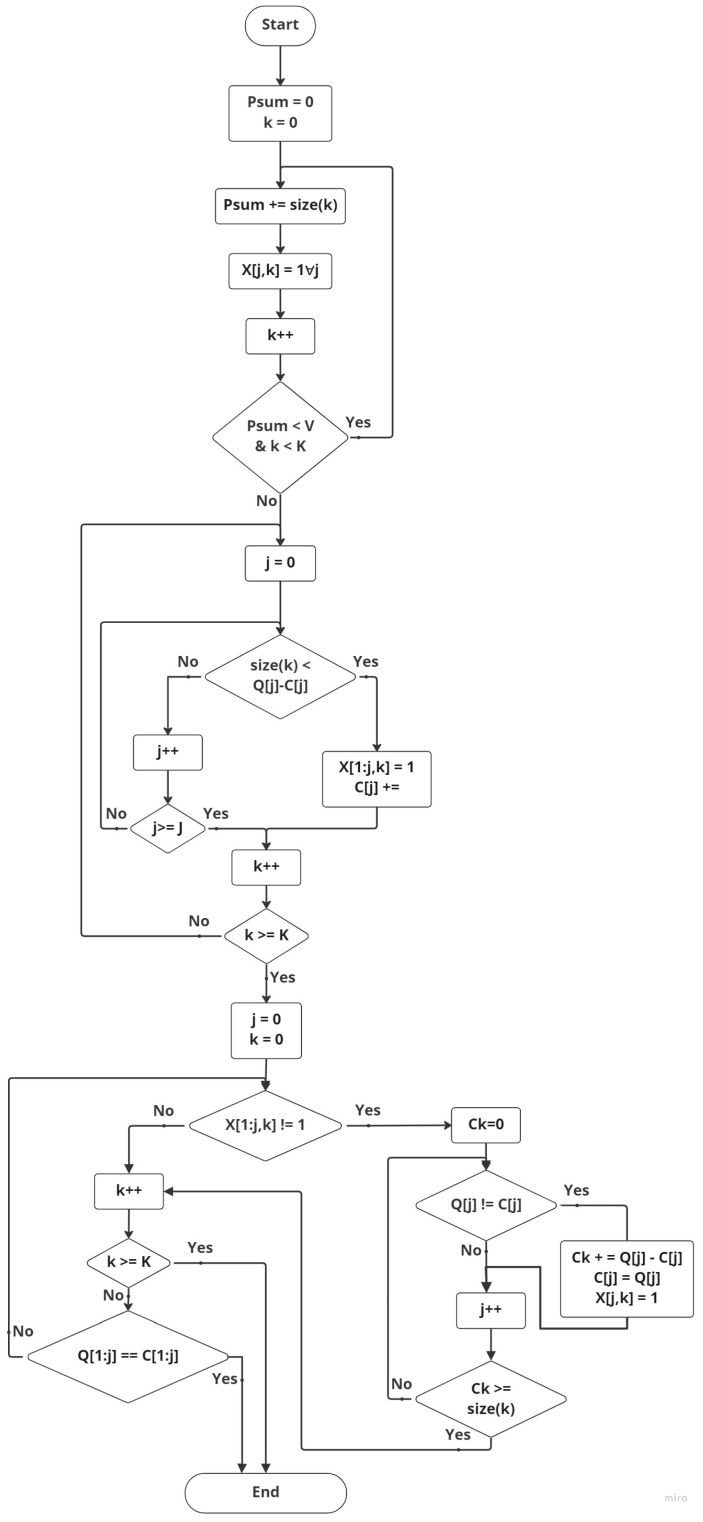
Flow chart of the proposed system model for the CMUMS Caching mechanism.

**Figure 5 sensors-23-00996-f005:**
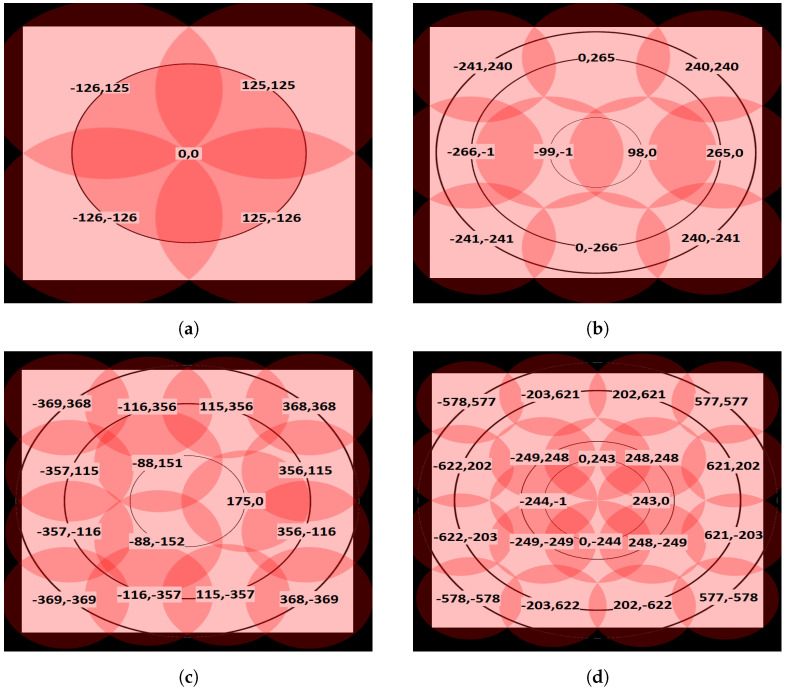
The estimated cluster formatted coverage for the four topology structures using Unity. (**a**) 5 MECs allocation in 500 × 500 m using unity; (**b**) 10 MECs allocation in 700 × 700 m using unity; (**c**) 15 MECs allocation in 1000 × 1000 using unity; (**d**) 20 MECs allocation in 1500 × 1500 using unity.

**Figure 6 sensors-23-00996-f006:**
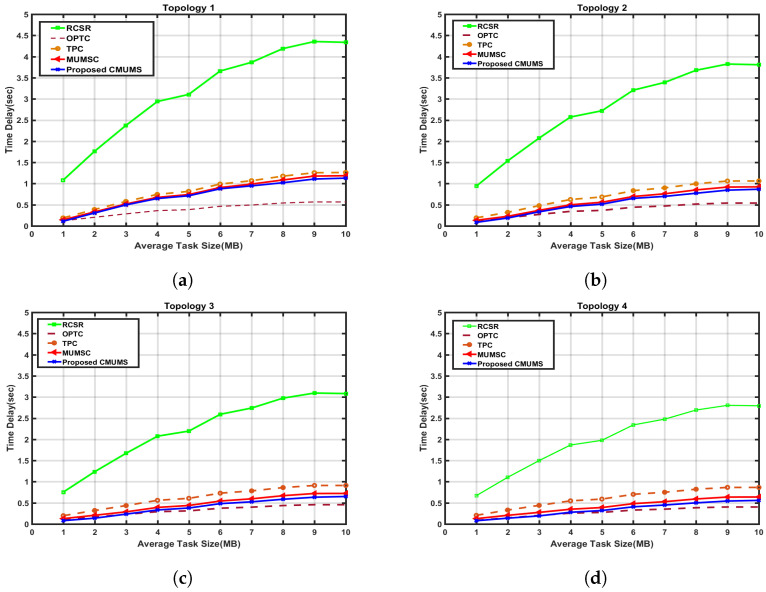
Time Delay vs. Average Task Size in four topology structures. (**a**) Time delay vs. Average task in 1st topology; (**b**) Time delay vs. Average task in 2nd topology; (**c**) Time delay vs. Average task in 3rd topology; (**d**) Time delay vs. Average task in 4th topology.

**Figure 7 sensors-23-00996-f007:**
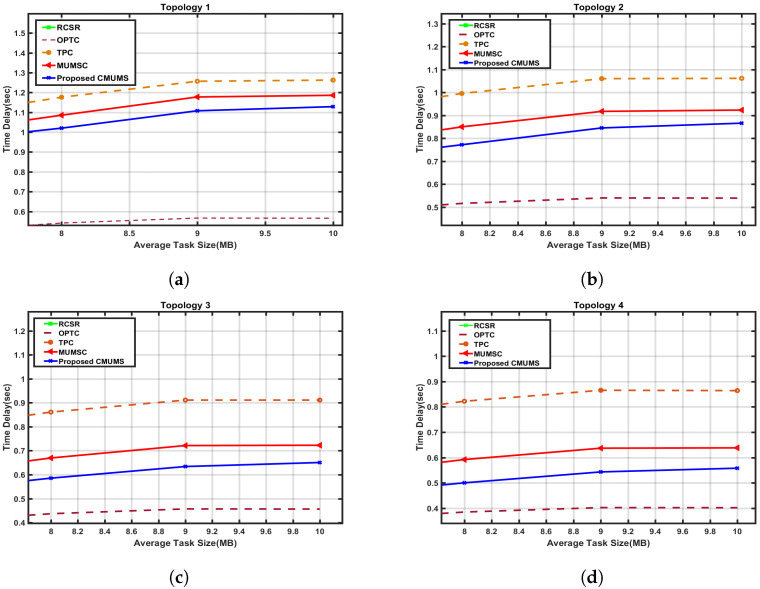
Cross-section of Time Delay vs. Average Task Size in four topology structure. (**a**) Time delay vs. Average task in 1st topology; (**b**) Time delay vs. Average task in 2nd topology; (**c**) Time delay vs. Average task in 3rd topology; (**d**) Time delay vs. Average task in 4th topology.

**Figure 8 sensors-23-00996-f008:**
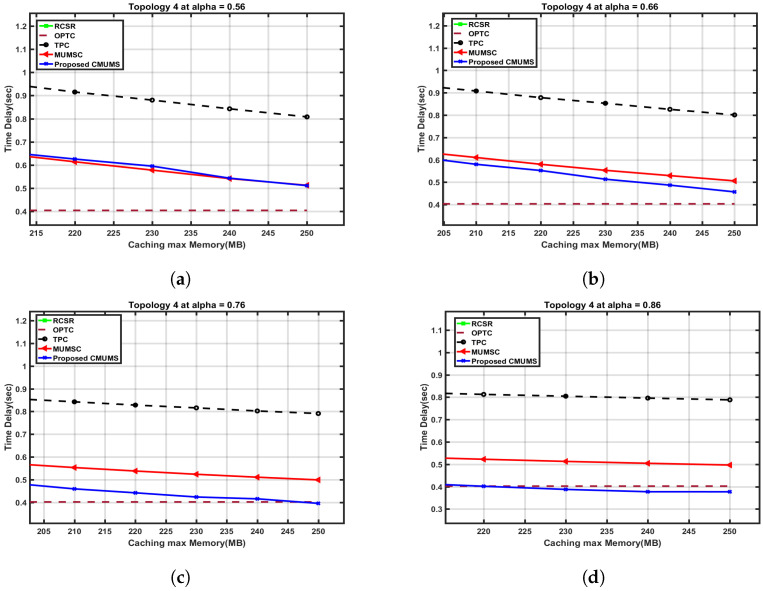
Cross-section of Time Delay vs. Q up to 250 MB in topology 4 at different α. (**a**) Time delay vs. Q up to 250 MB; (**b**) Time delay vs. Q up to 250 MB; (**c**) Time delay vs. Q up to 250 MB; (**d**) Time delay vs. Q up to 250 MB.

**Figure 9 sensors-23-00996-f009:**
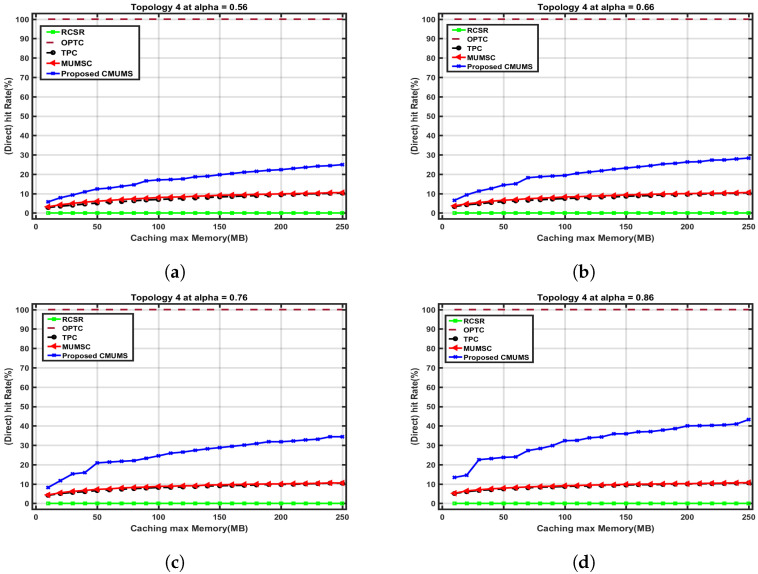
Direct hit rate vs. caching max Memory (Q) up to 250 MB at different α. (**a**) Direct hit rate vs. Q (at α = 0.56 as per [21]); (**b**) Direct hit rate vs. Q at α = 0.66; (**c**) Direct hit rate vs. Q at α = 0.76; (**d**) Direct hit rate vs. Q at α = 0.86.

**Figure 10 sensors-23-00996-f010:**
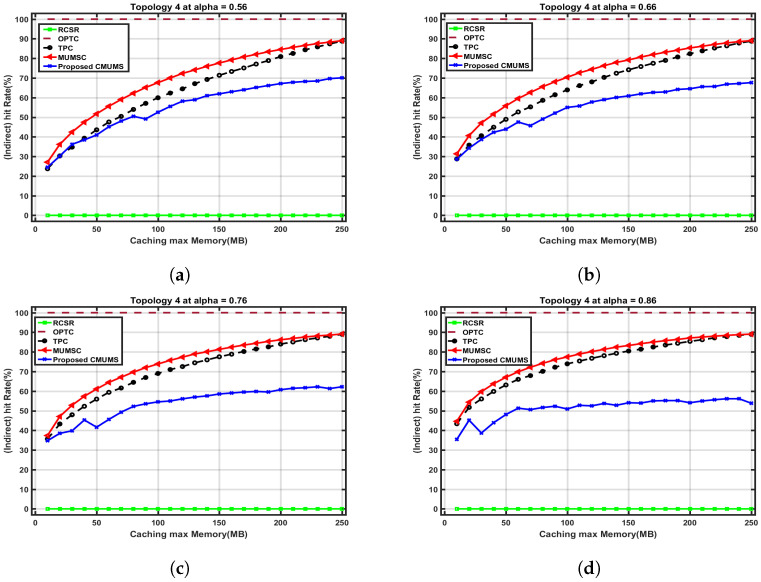
Indirect hit rate vs. caching max Memory (Q) up to 250 MB at different α. (**a**) Indirect hit rate vs. Q (at α = 0.56 as per [21]); (**b**) Indirect hit rate vs. Q at α = 0.66; (**c**) Indirect hit rate vs. Q at α = 0.76; (**d**) Indirect hit rate vs. Q at α = 0.86.

**Figure 11 sensors-23-00996-f011:**
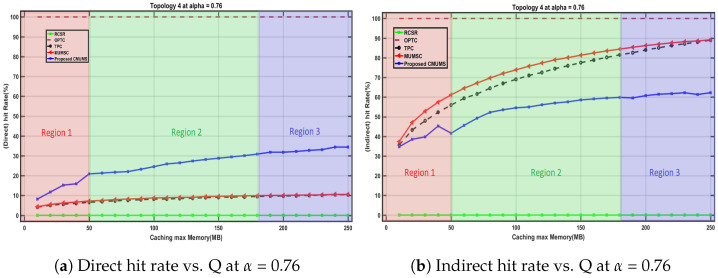
Direct and Indirect hit rate behavior vs. Q.

**Figure 12 sensors-23-00996-f012:**
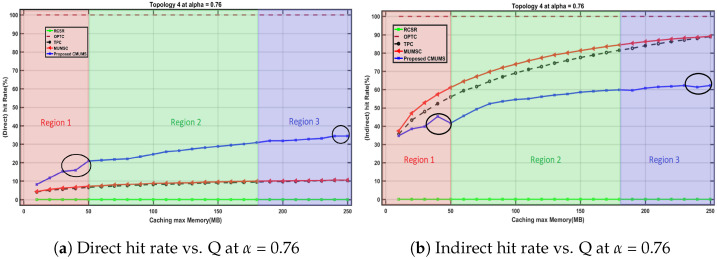
Direct and Indirect hit rate behavior vs. Q.

**Figure 13 sensors-23-00996-f013:**
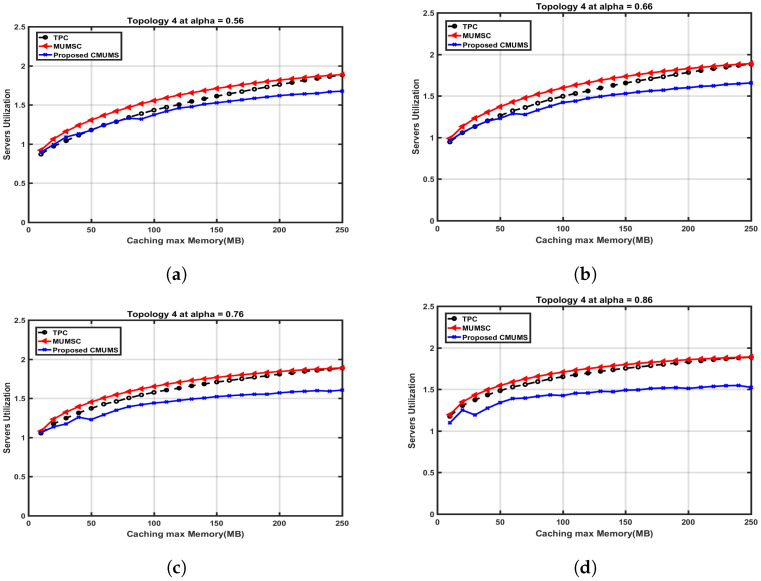
Server processor utilization ratio vs. caching max Memory (Q) up to 250 MB at different α. (**a**) Server utilization ratio vs. Q (at α = 0.56 as per [21]); (**b**) Server utilization ratio vs. Q at α = 0.66; (**c**) Server utilization ratio vs. Q at α = 0.76; (**d**) Server utilization ratio vs. Q at α = 0.86.

**Figure 14 sensors-23-00996-f014:**
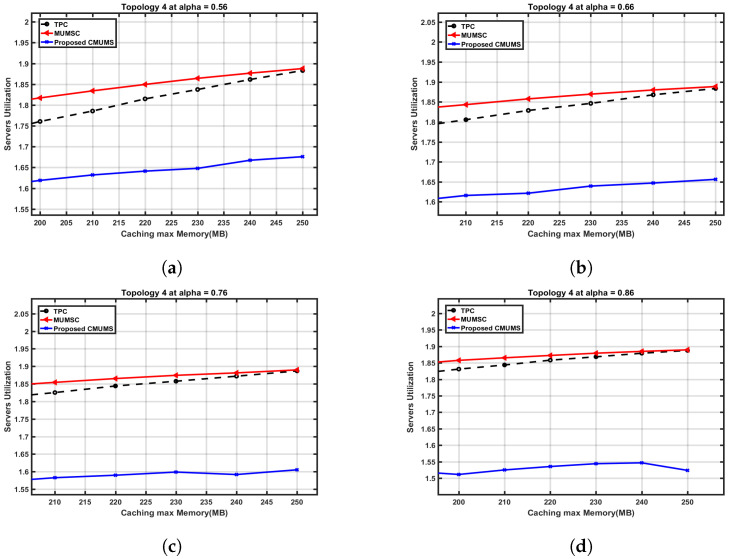
Cross section of server occupancy utilization ratio vs. caching max Memory (Q) up to 250 MB at different α. (**a**) Server utilization ratio vs. Q (at α = 0.56 as per [21]); (**b**) Server utilization ratio vs. Q at α = 0.66; (**c**) Server utilization ratio vs. Q at α = 0.76; (**d**) Server utilization ratio vs. Q at α = 0.86.

**Figure 15 sensors-23-00996-f015:**
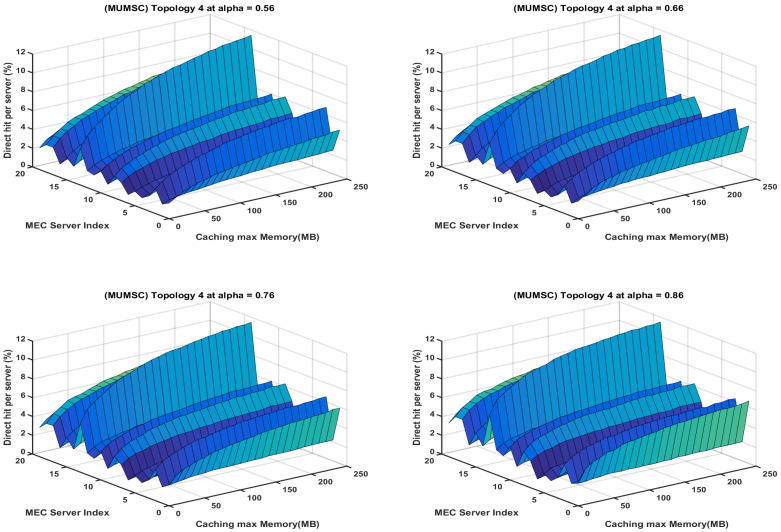
Heatmap represents direct hit rate vs. caching max Memory per each server for the MUMSC algorithm at different α.

**Figure 16 sensors-23-00996-f016:**
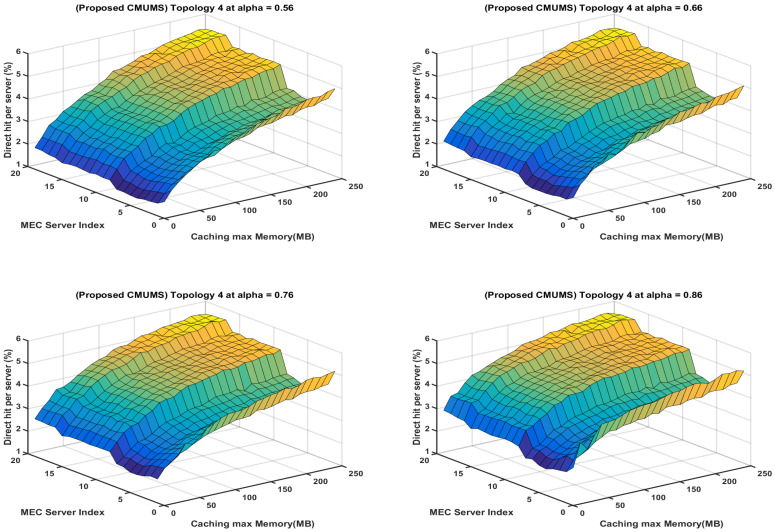
Heatmap represents direct hitrate vs. caching max Memory per each server for the CMUMS caching algorithm at different α.

**Table 1 sensors-23-00996-t001:** List of Symbols and Acronyms.

Symbol	Acronyms
N	Set of MDs (Mobile Devices).
*S*	Set of MEC servers (Mobile Edge Cloud Servers).
$S_{i}$	Set of Mobile Edge Cloud Servers that connects to MD *i*.
*K*	Set of computation tasks.
$\rho_{k}$	The amount of the task result to the whole size of task *k*.
$\beta_{k}$	The ratio of task result to the whole size of task *k*.
$p_{k}$	The popularity of task *k*.
*Q*	The caching space of mobile edge cloud servers.
$\alpha_{i,j}$	The connection probability between
	mobile devices *i* and server *j*.
$x_{j}^{k}$	If server *j* caches task *k*, $x_{j}^{k}$ = 1, otherwise $x_{j}^{k}$ = 0.
$t_{i,j}^{k}$	Delay in transporting result of task *k* between
	mobile device *i* and server *j* directly.
$t_{i,\alpha ,j}^{k}$	Delay in transporting result of task *k* between
	mobile device *i* and the agent server plus agent server and server *j*.
$t_{i,c}^{k}$	Delay in transporting result of task *k* between
	mobile device *i* and the remote center cloud server.
$p^{t}$	Transmit power in mWatt

**Table 2 sensors-23-00996-t002:** System parameters’ summary.

Parameter	Value
Topology sizes	500 × 500, 700 × 700, 1000 × 1000, 1500 × 1500
$f_{c}$	1 GHz
*B*	20 MHz
*Q*	100 MB & 250 MB
$p^{t}$	20 mWatt
σ	$4\times 10^{-3}$ mWatt
α	0.56, 0.65, 0.76, 0.87
$t_{max}$	1 s
$t_{dmax}$	2 s
$\rho_{k}$	(10, 2)
$\beta_{k}$	(0.5, 0.15)

**Table 3 sensors-23-00996-t003:** Time Delay vs. Average task size at different α for comparing the referenced algorithms in [21] with CMUMS at *Q* = 250 MB.

Time Delay for Algorithms	at α = 0.16	at α = 0.26	at α = 0.36	at α = 0.46	at α = 0.56	at α = 0.66	at α = 0.76	at α = 0.86
RCSR [21]	2.8896	2.8866	2.8829	2.8886	2.8903	2.8914	2.9006	2.9096
OPTC [21]	0.4179	0.4175	0.4168	0.4177	0.418	0.418	0.4194	0.4207
TPC [21]	0.9318	0.902	0.8582	0.8477	0.8301	0.8242	0.8223	0.8212
MUMSC [21]	0.6059	0.5829	0.552	0.5433	0.5304	0.5243	0.5216	0.5196
CMUMS	0.7174	0.6383	0.527	0.4897	0.4442	0.4109	0.3923	0.3733

**Table 4 sensors-23-00996-t004:** Direct hit rate at different α for comparing the referenced algorithms in [21] with CMUMS at *Q* = 250 MB.

Direct Hit Rate for Algorithms	at α = 0.16	at α = 0.26	at α = 0.36	at α = 0.46	α = 0.56	at α = 0.66	at α = 0.76	at α = 0.86
RCSR [21]	0	0	0	0	0	0	0	0
OPTC [21]	100	100	100	100	100	100	100	100
TPC [21]	9.6129	9.7308	9.8904	10.0075	10.0242	10.1658	10.2817	10.2817
MUMSC [21]	10.0913	10.2688	10.2175	10.3646	10.3154	10.3492	10.4846	10.4658
CMUMS	17.4467	18.0967	18.9462	20.6521	23.4821	26.7892	31.8783	37.94

**Table 5 sensors-23-00996-t005:** Indirect hit rate at different α for comparing the referenced algorithms in [21] with CMUMS at *Q* = 250 MB.

Indirect Hit Rate for Algorithms	at α = 0.16	at α = 0.26	at α = 0.36	at α = 0.46	α = 0.56	at α = 0.66	at α = 0.76	at α = 0.86
RCSR [21]	0	0	0	0	0	0	0	0
OPTC [21]	100	100	100	100	100	100	100	100
TPC [21]	84.8525	85.425	85.8467	86.3504	86.965	87.4229	87.7921	88.2492
MUMSC [21]	87.45	87.5763	87.87	87.9583	88.2179	88.4854	88.5829	88.8079
CMUMS	72.77	73.1521	72.9575	72.1592	69.6117	67.9796	63.7858	58.7708

**Table 6 sensors-23-00996-t006:** Summary of our proposed CMUMS model advantages.

Algorithm	Time Delay (s)		Direct Hit Rate Ratio		Indirect Hit Rate Ratio	
	at α = 0.56	at α = 0.86	at α = 0.56	at α = 0.86	at α = 0.56	at α = 0.86
MUMSC [21]	0.5304	0.5196	10.3154	10.4658	88.2179	88.8079
Proposed CMUMS	0.4442	0.3733	23.4821	37.94	69.6117	58.7708

## Data Availability

Not applicable.

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
