# Peer review of "Cluster-Based Multi-User Multi-Server Caching Mechanism in Beyond 5G/6G MEC"

_sensors, 2023, doi:10.3390/s23020996_

Round 1

Reviewer 1 Report

This paper proposed Cluster-based Multi-User Multi-Server cashing algorithm to optimize the content caching mechanism in Mobile Edge Computing. The optimized caching algorithm proposed by the authors addressed the problems of network optimization in addition to the related end-user quality of experience by reducing the latency and increasing the direct hit rate. And through the simulation experiment, it is compared with other system models to prove its superior performance. This paper is well worth reading. However, I have the following comments.

(1) In the introduction of the paper, the authors should focus on the technical difficulties MEC faces today, point out what aspects of the solution he has proposed have been optimized, and finally explain their contribution to the implementation of the solution.

(2) The relevant literature involved in the second part of the paper is too comprehensive and has no focus. The authors should explain the key advantages of the work compared with these literature.

(3) Please explain that in Figure 14 (d), with the increase of caching max Memory, the server occupancy utilization ratio of CMUMS has a downward trend.

(4) It is suggested to add more recent MEC references in the introduction, e.g., Online Edge Learning Offloading and Resource Management for UAV-Assisted MEC Secure Communications, IEEE Journal of Selected Topics in Signal Processing, DOI, 10.1109/JSTSP.2022.3222910, 2022. Secure transmission for multi-UAV-assisted mobile edge computing based on reinforcement learning, IEEE Transactions on Network Science and Engineering, DOI:10.1109/TNSE.2022.3185130, 2022.

(5) The authors should proof read the paper to further improve the writing.

Reviewer 2 Report

Review Attached

Author Response

We would like to thank you for the time and effort you spent evaluating our article and providing comments.

Reviewer 3 Report

In this work, authors studied the service caching and the task offloading for mobile edge computing. This is a hot and promising research directions, due to the development of ICT. In my opinion, there are some issues should be addressed for their work.

1. Abstract should be improved by focusing on the innovation and contribution of their work instead of the background.

2. The presentation of background is too verbose in Introduction, which should be improved. And the first two paragraphs are too long.

3. The innovation and contribution of their work should be highlighted in Introduction.

4. The second section should be named as Related Works. Strengths and weaknesses of each related work should be discussed. Works published recently should be included.

5. How the four different topology structures are generated should be illustrated.

6. The presentation of their methods should be improved, as the current version is hard to be understood.

7. Authors should present the model, i.e., the problem statement / formulation, before their methods.

8. The baseline methods should be introduced briefly in their experiments.

9. How are the task loads generated for the experiments?

10. The weakness of their work should be discussed in the Conclusions section.

11. The content should be improved in logic, in overall. The current form of the manuscript make the work hard to be understood.

Reviewer 4 Report

This paper proposes a cashing algorithm to optimize the content caching mechanism and control distributing the high-probability tasks. The work attempts to addresses the integer optimization problem that controls which content will be cached and the hosting server list in order to optimize the available resources utilization and develop a completely new level of services and innovative approaches.

Overall, the idea is clear and interesting, and the paper is well written. In addition, most of the figures are aesthetically standard. Nevertheless, there are a few major concerns.

1. In the abstract, there is too much background and not enough description of the contributions and experimental results. Please revise the abstract of the paper, and highlight the main contribution and results. 

2. The first paragraph in the Section Introduction is too long. It is recommended to break it up into several paragraphs according to the relationship and highlight the main ideas. 

3. Figure 4 does not have enough resolution and needs to be redrawn.

4. In the relevant standards of 5G, the full name of MEC is multiple-access edge computing. It is recommended to modify the full name of the MEC.

Round 2

Reviewer 3 Report

The manuscript has been sufficiently improved.